# A single amino acid variant in the variable region I of AAV capsid confers liver detargeting

Ruxiao Xing[1‡], Mengyao Xu[1‡], Darcy Reil[1], April Destefano[1], Mengtian Cui[1], Nan Liu[1], Jialing Liang[1], Guangchao Xu[1], Li Luo[1], Meiyu Xu[1], Fang Zhang[1], Phillip W.L. Tai[1], Yuquan Wei[2], Alisha M. Gruntman[1], Terence R. Flotte[1], Guangping Gao[1,3]*, Dan Wang [1,4]*

**1** Department of Genetic and Cellular Medicine, Horae Gene Therapy Center, University of Massachusetts Chan Medical School, Worcester, Massachusetts, United States of America, **2** State Key Laboratory of Biotherapy, West China Hospital, Sichuan University, Chengdu, China, **3** Department of Microbiology, University of Massachusetts Chan Medical School, Worcester, Massachusetts, United States of America, **4** RNA Therapeutics Institute, University of Massachusetts Chan Medical School, Worcester, Massachusetts, United States of America

‡ Co-first authors.
* Dan.Wang@umassmed.edu (DW); Guangping.Gao@umassmed.edu (GG)

## Abstract

AAV capsid serotypes isolated from nature have been widely used in gene delivery and gene therapy. Recently, more than 1,000 distinct AAV capsids were identified from human clinical samples by high-throughput, long-read DNA sequencing. In this study, we tap into this broad natural biodiversity of AAV capsids to develop liver-tropic AAV capsids. We initially screened a subset of variants derived from AAV8 (n = 159) for packaging efficiency. The high-yielding variants were subjected to a barcoded vector library screen in mice and ferrets for their ability to mediate liver gene transfer. Although no variant surpassed AAV8 for liver targeting, several exhibited a liver detargeting phenotype. Among these, we focused on the N271D variant (AAV8 VP1 numbering), located in the variable region I (VR-1), which has been previously implicated in influencing liver tropism. The liver detargeting phenotype of AAV8.N271D was confirmed by single vector administration in mice. Additionally, we grafted the N271D variant onto AAV9 and MyoAAV capsids (N270D by AAV9 VP1 numbering). The AAV9.N270D and MyoAAV.N270D vectors showed a similar liver-detargeting phenotype, although muscle targeting was moderately reduced. Although we did not identify any capsid variants that outperform AAV8 in liver transduction, this study reinforces the important role of VR-1 in modulating liver tropism and highlights the potential of engineering VR-1 residues to reduce liver gene transfer and associated toxicity observed in several gene therapy studies.

**Data availability statement:** Nanopore sequencing and Illumina sequencing data are deposited in NCBI Sequence Read Archive (SRA) and available with accession ID PRJNA1268899.

**Funding:** This study was supported by a grant from the National Institutes of Health (NIH, https://www.nih.gov) (P01HL158506 to D.W.). The funder did not play any role in the study design, data collection and analysis, decision to publish, or preparation of the manuscript.

**Competing interests:** I have read the journal's policy and the authors of this manuscript have the following competing interests: G.X., L.L., P.W.L.T., Y.W., G.G. are inventors of a patent application regarding the AAV8 variants described in this study (US20230374545A1). G.G. is a scientific co-founder of Voyager Therapeutics, Adrenas Therapeutics and Aspa Therapeutics and holds equity in these companies.

## Author summary

Adeno-associated virus (AAV) is widely used as a delivery vehicle for gene therapy. Its capsid is composed of 60 viral protein subunits and largely determines key features such as tissue tropism and host immune responses. Many naturally occurring AAV capsids have been isolated and now serve as the workhorses in the gene therapy field. In this study, we screened a diverse panel of AAV capsid variants previously identified from human clinical samples to discover candidates with enhanced liver tropism. In a high-throughput comparative analysis, although no capsid variant outperformed the parental AAV8 capsid in targeting the mouse liver, one variant exhibited markedly reduced liver tropism despite differing from AAV8 by only a single amino acid in the capsid protein. Further investigation demonstrated that introducing this single amino acid change into other AAV capsids similarly reduced their liver targeting. Notably, this amino acid resides in a structural region previously implicated in modulating liver tropism. These findings suggest a rational strategy to engineer liver-detargeting AAV vectors, potentially reducing liver-associated toxicity in systemic gene therapy applications targeting non-hepatic tissues.

## Introduction

Adeno-associated virus (AAV) has a single-stranded DNA genome packaged in an icosahedral capsid that consists of 60 protein monomers, VP1, VP2, and VP3 [1]. The capsid initiates interactions with the host, including binding to cell surface receptors and recognition by the host immune system. In recombinant AAV (rAAV), which is widely used as an *in vivo* gene therapy delivery platform [2], the capsid is either identical to or derived from wild-type AAV (wtAAV), and therefore is inherently critical to determine therapeutically relevant properties, including tissue tropism and immunogenicity. To develop AAV capsids with clinically favorable properties, one way is to isolate wtAAV capsid sequences that exist in nature followed by testing in the context of rAAV. Naturally occurring wtAAV capsids evolve within host tissues during infections [3] and may inherently possess desirable features for therapeutic gene delivery, such as tissue-specific fitness and the ability to evade host immune surveillance. This approach has led to the discovery of a series of AAV capsid serotypes [4], many of which have been used in clinical gene therapy and continue to serve as the workhorse for approved AAV-based gene therapies to date [5]. Additionally, these naturally occurring AAV capsid variants serve as a foundation for capsid engineering through various approaches, most notably peptide insertion combined with library selection and directed evolution, to further enhance desirable rAAV attributes [6].

From a biology perspective, these naturally occurring AAV capsid sequences offer ample opportunities to study the sequence- and structure-function relationship. For example, aligning the amino acid sequences of various AAV capsids results in the identification of nine variable regions (VRs), which diverge to a

higher degree than the remaining regions among different serotypes. Structural studies revealed that VR-I to VR-IX are mostly exposed to the outer surface of the assembled capsid, and play critical roles in binding to a diverse array of cell surface receptors and host antibodies [7]. For example, AAV2.5 was created by incorporating five residues from AAV1 into AAV2, and showed enhanced muscle tropism [8]. A follow-up study demonstrated that, among the five residues, the 265T insertion in VR-I determined the high muscle tropism of AAV2.5 [9]. AAV-DJ is an engineered capsid created by shuffling eight capsid serotypes and exhibits resistance to neutralization by human intravenous immunoglobulin (IVIG) [10]. Structural studies showed that the unique structure of VR-1 in AAV-DJ as compared with AAV2 abrogated binding to the A20 neutralizing antibody [11]. Indeed, VR-1 contributes to the 3-fold protrusions on AAV capsid [12], which play critical roles in binding to cell surface receptors and antibodies [13]. Understanding the capsid-host interactions at the molecular level also enables rational engineering by manipulating specific residues to achieve desired capsid properties [12,14]. Recent high-throughput, long-read DNA sequencing technologies have allowed more efficient and in-depth profiling of naturally existing AAV capsid variants. For example, we previously identified more than 1,000 unique AAV capsid variants from human clinical biopsies using the PacBio sequencing platform [15]. An AAV2-derived variant, named AAVv66, exhibits enhanced production yields, virion stability, and central nervous system (CNS) transduction [15].

Two liver-targeted gene therapies delivered by AAV vectors have been approved for treating hemophilia A and hemophilia B, respectively. In both cases, therapeutic levels of the blood clotting factor VIII [16] or factor IX [17] secreted from the liver into the bloodstream could be attained. However, transducing the liver to function as a bio-factory for producing and secreting alpha-1 antitrypsin (A1AT), the second most abundant secreted serum protein [18], proved challenging in reaching the therapeutic threshold in the bloodstream for treating A1AT deficiency [19], suggesting that more efficient liver targeting may be required. Among the AAV capsid variants that we previously identified from human clinical samples [15], a large portion are AAV8-derived (i.e., exhibiting the highest degree of sequence homology to AAV8) [20]. Given AAV8's strong liver tropism, in this study, we tapped into this diverse repertoire of AAV8 variants to characterize their liver tropism in mice using a barcoded library screen approach, aiming to identify more potent liver-tropic capsid variants as potential vectors to deliver A1AT deficiency gene therapy. In addition, ferret models of A1AT deficiency were recently generated and characterized to be a platform for preclinical testing of therapeutics including gene therapy [21]. Therefore, we also screened the same capsid library in ferrets to investigate cross-species tropism of emerging variants. Although we found that no variants surpassed AAV8 for liver gene transfer, we identified one variant with a liver detargeting phenotype mediated by a single N271D residue change in VR-1 (AAV8 VP1 numbering). This study adds to the recent literature showing the important role of VR-1 in liver gene transfer [22–24]. Furthermore, several studies on systemic AAV vector administration have demonstrated liver toxicity [25–28], suggesting the potential safety benefit with a liver-detargeting capsid to deliver gene therapy to other organs, such as the CNS and muscle. The key VR-1 residues that influence liver gene transfer identified by us and others may provide a rational avenue to develop AAV vectors with reduced liver toxicity.

## Results

### Generation and characterization of an AAV8 vector library

We first constructed a total of 159 packaging plasmids, each expressing a unique AAV8 capsid variant identified from human tissues (S1 Table) along with the AAV2 Rep. These plasmids were used individually to package an *EGFP* transgene cassette in a small-scale AAV vector production assay to determine their vector production yield and to benchmark against the parental AAV8 capsid. Overall, the AAV8 capsid variants were less efficient than AAV8, exhibiting 63% or lower vector production yield (**Fig 1a**), indicating that the amino acid residue changes may compromise capsid assembly or vector genome packaging. We arbitrarily selected the top 37 variants for the subsequent library screen, as low vector production yield will pose a translational hurdle for a gene therapy delivery vehicle.

PLOS Pathogens

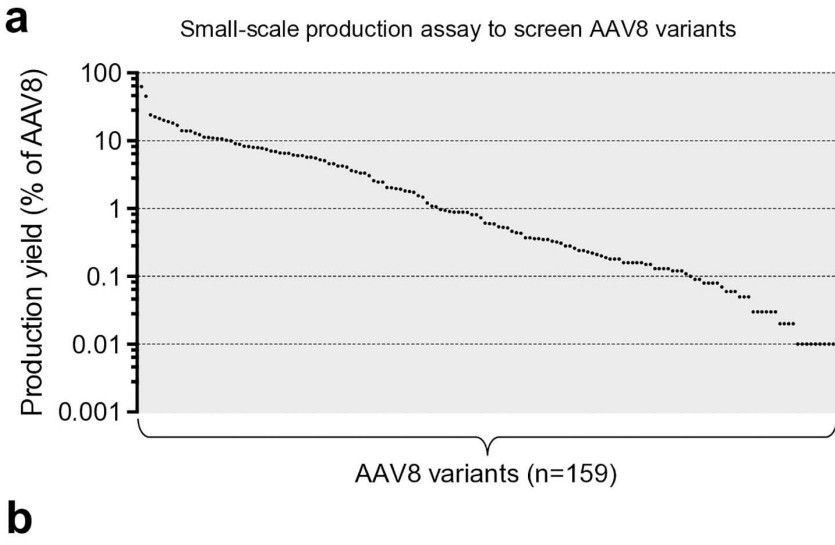

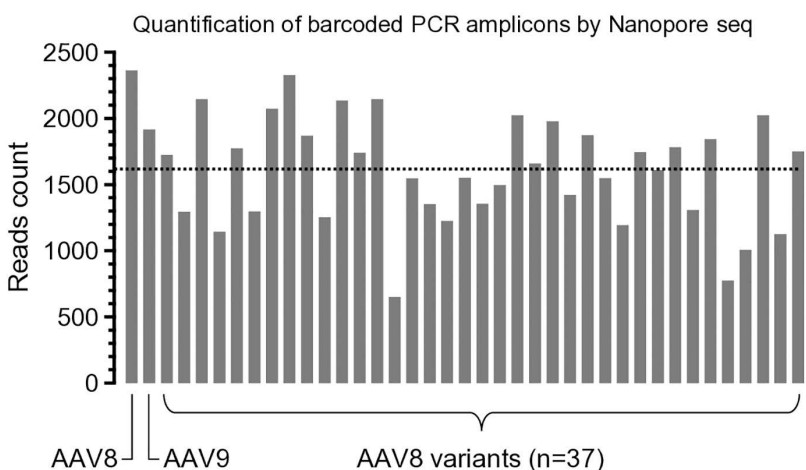

**Fig 1. Generation and characterization of the AAV8 vector library.** (a) Dot plot showing the production yield of each AAV8 capsid variant in a small-scale vector production assay packaging the same *EGFP* transgene cassette. Each dot represents a unique capsid variant. The production yield is normalized to the AAV8 vector level (defined as 100%) and ranked from the highest to the lowest. Data were based on one biological repeat. (b) Bar graph showing the count of Nanopore sequencing reads mapped to the unique vector transgene barcodes packaged in AAV8, AAV9, or AAV8 variants. The dashed line shows the mean value. Data were based on one biological repeat.

Next, we constructed a series of gene-of-interest (GOI) plasmids, each expressing a barcoded, non-coding Tough Decoy (TuD) RNA under the control of the U6 promoter as previously reported [29] (S1 Fig). We chose not to use a protein-coding reporter transgene, such as *EGFP,* to avoid potential immunogenicity when conducting the screen in large animals such as ferrets. A pair of unique packaging and GOI plasmids were used in large-scale AAV vector production, so that each barcoded *TuD* transgene represented a unique capsid. These vectors were individually purified, followed by vector genome (vg) titer determination. According to their vg titers, these vectors were then pooled at various volumes with the goal of equal representation for each capsid vector in the library. We extracted the pooled library vector DNA, amplified the barcode-containing region by PCR, and performed high-throughput nanopore sequencing to quantify the relative abundance of each barcode. This abundance served as a proxy for the relative representation of each capsid vector in the library. Although different barcode sequences may introduce PCR bias, we reasoned that using the same PCR method to

quantify vector abundance in both the library and tissue samples would mitigate this bias, provided that library abundance is used as a normalization control. We found that the abundance of the barcodes representing 39 capsid vectors (i.e., AAV8, AAV9, and the 37 AAV8 variants) deviated from the average by less than 2.5-fold (**Fig 1b**), comparable to other barcoded libraries generated in the same manner (approximately 7-fold) [30]. In contrast, purifying all capsid vectors in bulk would likely result in skewed representation in favor of "good producers" (e.g., more than 200-fold deviation from the average [30]), which may cause bias in subsequent functional screens. Alternatively, individual capsid titers could be adjusted at the crude lysate stage prior to pooling for bulk purification. However, we reasoned that purifying and storing individual vectors would provide greater flexibility for downstream applications, such as enabling the use of single vectors or selected subsets for specific tests.

### Vector library screen in mice and ferrets

The pooled vector library was delivered to three adult wild-type (WT) mice at $2 \times 10^{13}$ vg/kg via tail vein. Four weeks post-treatment, the mice were euthanized for tissue collection. The total DNA extracted from livers or the vector library was subjected to PCR to amplify the barcode-containing region in vector DNA. The barcodes present in the amplicons were quantified by Illumina sequencing and normalized to their relative abundance in the vector library. Importantly, the relative abundance of each barcoded amplicon in the library was consistent with the results obtained by nanopore sequencing (**Fig 1b**, **S2 Fig**). This analysis revealed that no candidate variants could surpass AAV8 for liver gene delivery (**Fig 2a**). In parallel, the same vector library was screened in three WT ferrets sero-negative for AAV8 and AAV9 (**S3 Fig**) in the same fashion, which led to the similar finding that AAV8 outperformed all candidate variants (**Fig 2b**). Notably, the candidate variants generally exhibited a similar trend of liver gene delivery efficiency in mice and ferrets (**Fig 2c**), suggesting the robustness of the screening pipeline. As several studies have shown that AAV vector DNA abundance does not necessarily correlate with transgene expression when using different AAV capsids to deliver the same cargo [31–33], we also quantified the barcoded *TuD* RNA levels in mouse livers. However, consistent with the vector DNA analysis, AAV8 outperformed all variants in terms of transgene expression (**Fig 3a**, blue bars).

### Liver detargeting by several capsid variants

In both mice and ferrets, we consistently observed that several capsid variants, including v2, v5, v12, v13, v15, v23, v25, and v36, showed very low or barely detectable levels of liver gene delivery (**Fig 2a** and **2b**). As expected, these candidate variants also led to low or barely detectable *TuD* RNA levels in the mouse livers (**Fig 3a**). To investigate whether these capsid variants lost liver-specific or pan-tissue tropism, we quantified the *TuD* RNA levels in the heart and tibialis anterior (TA) muscle from the treated mice. Taken together, the RNA analysis revealed that several variants, including v2, v5, v23, v25, and v36, showed a strong liver detargeting phenotype, albeit with moderate reductions in *TuD* expression in the heart and TA muscle. In contrast, transduction by v12, v13, and v15 failed in all three tissue types (**Fig 3a**). We were not able to detect *TuD* transcripts in ferrets tissues, likely due to the overall low gene delivery efficiency in ferrets as compared to that in mice (**S4 Fig**).

The variants v2, v5, v23, v25, and v36 carry unique amino acid changes across the entire capsid protein (**S2 Table**). Among those, the N271D (AAV8 VP1 numbering) residue change in v5 is located in VR-1 (**S2 Table**), which had been implicated in modulating liver tropism in recent publications [22–24] (**Fig 3b**) (also see Discussion). Therefore, we focused on this variant for further validation by single vector treatment (**S5 Fig**).

### Validation of liver detargeting in mice by single vector administration

To further characterize the impact of N271D on tissue tropism in mice, we generated a pair of AAV8 and AAV8.N271D vectors packaging the same *EGFP* transgene cassette. As expected from the small-scale rAAV productivity assay

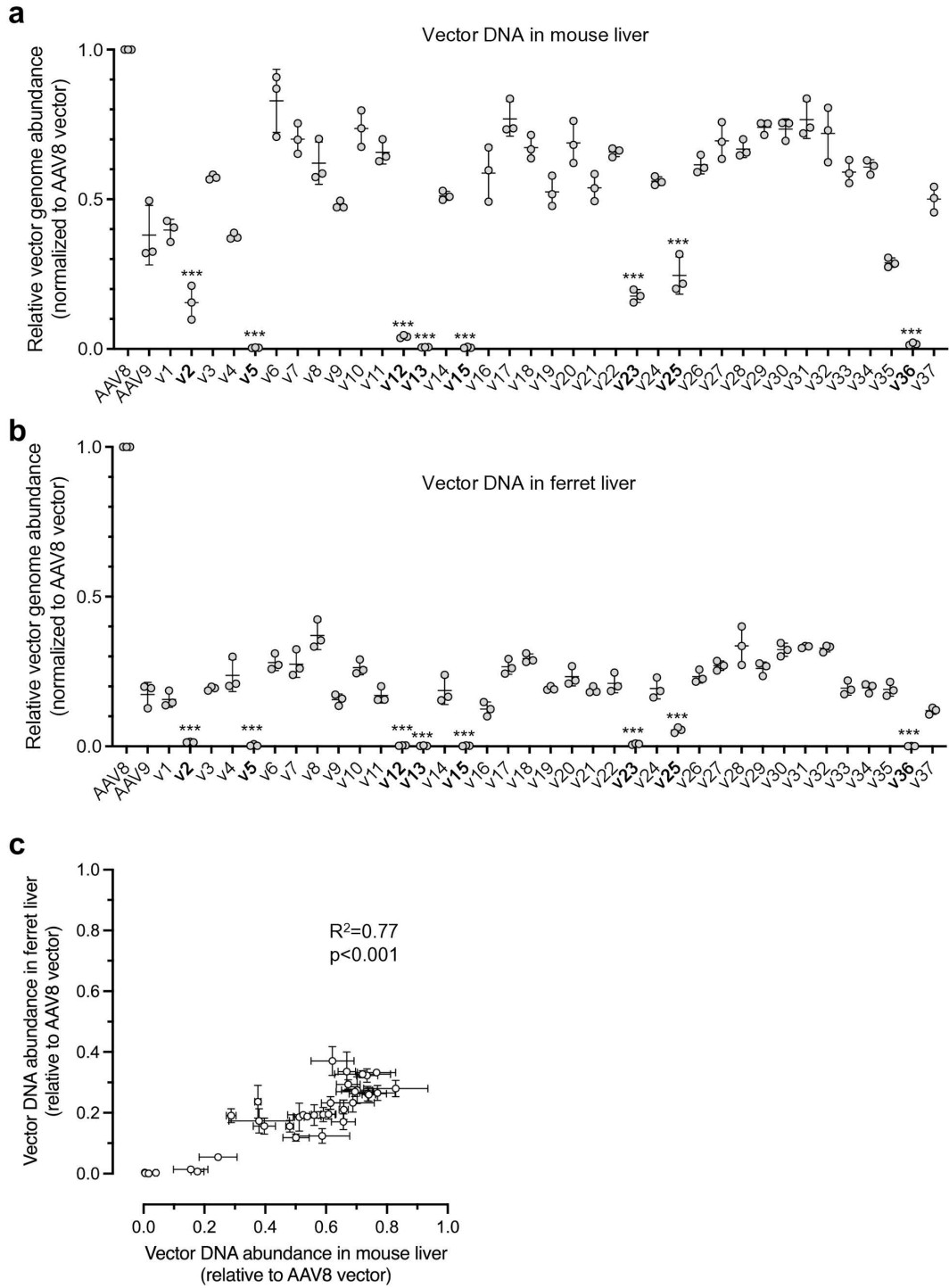

**Fig 2. Vector library screen in mice and ferrets.** (a, b) Scatter dot plots showing the relative vector genome abundance of each barcoded capsid vector in mouse (a) or ferret (b) livers. Data are normalized to the AAV8 vector level (defined as 1.0), and presented as mean ± SD. Each dot represents an individual animal. v1 to v37 denote the identifiers of AAV8 variants. The variants in bold indicate that they resulted in low or barely detectable levels of vector DNA. (c) Scatter plot showing the relationship between the relative vector DNA abundance of each capsid variant in mouse liver (x-axis) and ferret liver (y-axis). Data are presented as mean and standard deviation of three animals. In (a) and (b), statistical analysis is performed using two-tailed one-way ANOVA followed by Dunnett's multiple comparisons test against AAV8. ***$p < 0.001$. In (c), the linear regression statistics are shown.

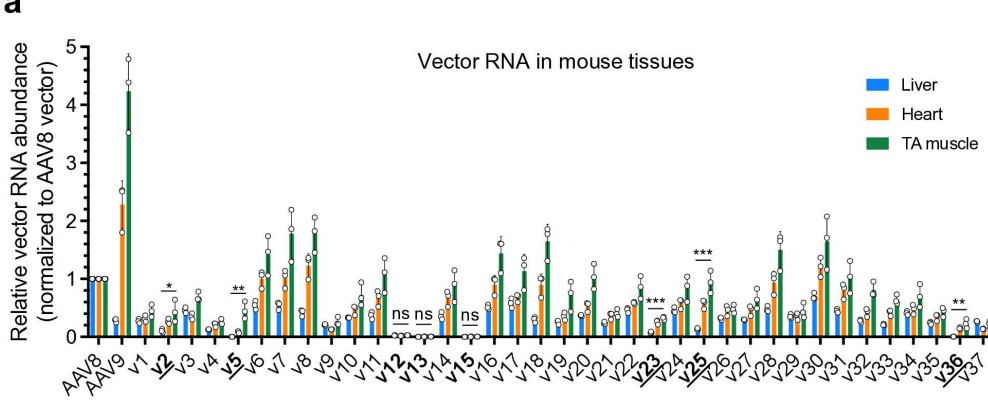

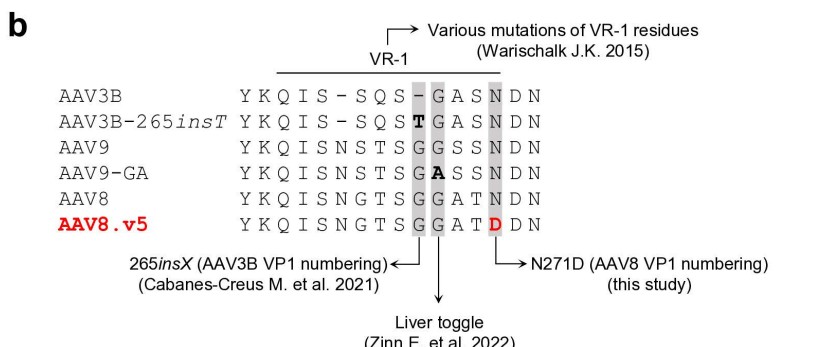

**Fig 3. AAV8.v5 shows liver detargeting phenotype in mice.** (a) Bar graph showing the relative vector RNA (cDNA) abundance of each barcoded capsid vector in mouse liver (blue), heart (orange), and tibialis anterior (TA) muscle (green). Data are normalized to the AAV8 vector level in respective tissues (defined as 1), and presented as mean±SD. Each dot represents an individual animal. v1 to v37 denote the identifiers of AAV8 variants. The variants in bold indicate that they resulted in low or barely detectable levels of vector RNA expression in the liver, with the ones underlined indicating well-detectable levels of vector RNA expression in the heart and/or TA muscle. Statistical analysis is performed using two-tailed one-way ANOVA to compare vector RNA levels in the three tissues transduced by each vector. *p<0.05; **p<0.01; ***p<0.001; ns: not significant. (b) Alignment of the amino acid sequences of multiple AAV capsids. Only variable region I (VR-1) and surrounding residues are shown as single-letter abbreviations. Dashes indicate gaps. The 265insT (AAV3B VP1 numbering) residue described in Cabanes-Creus M. et al. 2021 [22], the liver toggle residue described in Zinn E. et al. 2022 [23], and the N271D (AAV8 VP1 numbering) residue described in this study are highlighted with gray background. The residues that differ from parental capsids described in these studies are highlighted in bold. The Warischalk study [24] investigated a wide range of VR-1 mutants across multiple serotype capsids.

(Fig 1a), AAV8.N271D produced vectors less efficiently than the parental AAV8 capsid (S6 Fig). We treated two groups of mice with the two vectors at the same dose of $3 \times 10^{11}$ vg/mouse, respectively. Four weeks post-treatment, the mice were euthanized to compare vector DNA abundance, *EGFP* mRNA levels, and EGFP protein levels in the liver, heart, and TA muscle. Consistent with the library screen results, the AAV8.N271D.EGFP vector showed >100-fold reductions in vector DNA abundance and transgene expression in the liver as compared to the AAV8.EGFP vector, while heart and TA muscle targeting was moderately impacted (Fig 4a, S7a Fig).

As N271 (AAV8 VP1 numbering) is highly conserved among multiple serotypes (Fig 3b), we tested whether grafting the N271D residue change to other AAV capsids could confer liver detargeting. To this end, we introduced the homologous N270D (AAV9 VP1 numbering) mutation into the AAV9 capsid and generated the AAV9.N270D.EGFP vector. Following administration to WT mice, we observed a dramatic liver detargeting phenotype as compared to the parental AAV9.EGFP vector (Fig 4b, S7b Fig). Regarding transducing the heart and TA muscle, N270D in AAV9 capsid had a larger negative impact than N271D in AAV8 (compare the middle panels in Fig 4a and 4b). MyoAAV is an engineered capsid with a 7-mer

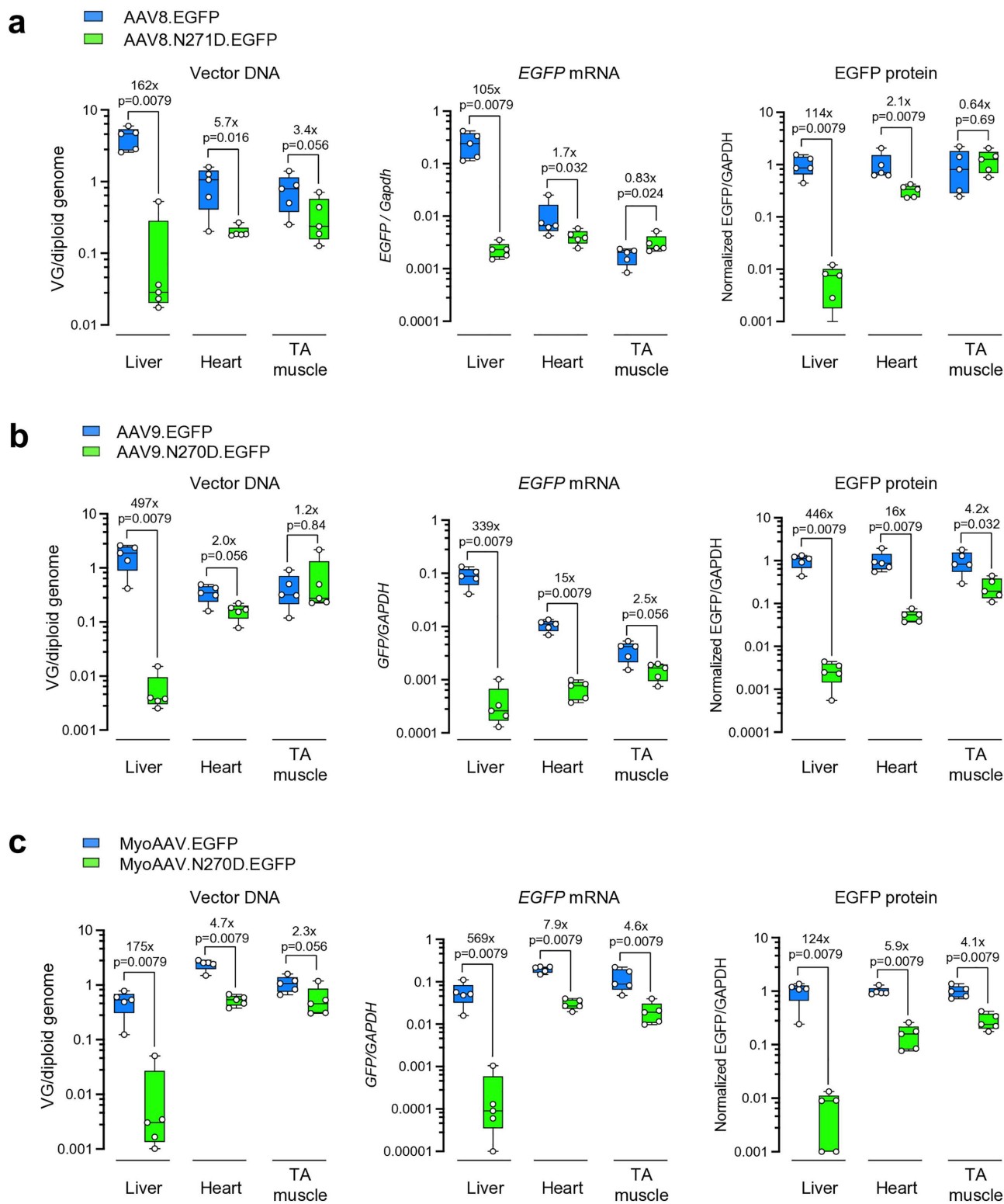

**Fig 4. Characterization of vector performance in mice following individual vector administration.** (a-c) Box plots showing the vector DNA abundance (left), *EGFP* mRNA levels (middle), and relative EGFP protein levels (right) in the liver, heart, and tibialis anterior (TA) muscle collected from the mice treated with AAV8 or AAV8.N271D vectors (a), AAV9 or AAV9.N270D vectors (b), and MyoAAV or MyoAAV.N270D vectors (c). Note that the protein levels are relative to the parental capsid in each tissue. Each dot represents an individual mouse. The box extends from the first to the third quartiles with

the line inside denoting median. The whiskers end at minimum and maximum values. The fold changes of medians and p values are labeled. Statistical analysis is performed using two-tailed non-parametric Mann-Whitney U test.

peptide insertion in AAV9 capsid proteins, and exhibits enhanced muscle tropism as compared with AAV9 in mice and monkeys [34]. To test whether the enhanced muscle tropism of MyoAAV can compensate for the reduced muscle targeting by AAV9.N270D, we generated a pair of MyoAAV.EGFP and MyoAAV.N270D.EGFP vectors and administered to mice. We consistently observed liver detargeting by MyoAAV.N270D in mice (Fig 4c, S7c Fig). Although MyoAAV.N270D. EGFP led to lower transduction levels in the heart and TA muscle as compared to MyoAAV.EGFP, it still outperformed AAV8.EGFP and AAV9.EGFP at the mRNA level (compare the middle panels of Fig 4a–4c). As our initial western blotting analysis compared each pair of parental and mutant capsids individually (S7a–S7c Fig), we subsequently performed a side-by-side western blotting to directly compare EGFP protein expression mediated by the AAV8, AAV9 and MyoAAV. N270D vectors. Consistent with mRNA quantification results, MyoAAV.N270D.EGFP expressed the highest EGFP protein levels in the heart and TA muscle relative to AAV8.EGFP and AAV9.EGFP (S7d Fig). Taken together, the N271D (AAV8 VP1 numbering) variant consistently conferred liver detargeting in the context of all three parental capsids tested (S8 Fig).

## Discussion

Residues in VR-1 have been shown to influence liver tropism of AAV vectors. Warischalk tested several VR-1 amino acid deletion mutants across multiple serotype capsids, and found that some detargeted from transducing mouse liver by orders of magnitude [24]. Using a domain swapping strategy, Cabanes-Creus et al. found that a single threonine insertion downstream of residue 264 of AAV3B (AAV3B-265insT, Fig 3b) greatly enhanced liver targeting in mouse [22]. In another study, Zinn et al. identified residue 267 (AAV9 VP1 numbering), named "liver toggle", as a key determinant for liver targeting [23] (Fig 3b). For example, a single G267A mutation in the AAV9 capsid (AAV9-GA) resulted in significant liver detargeting with respect to both vector genomes (1,522-fold) and expression (363-fold), with no significant difference in terms of gene transfer to the heart and quadriceps, and a moderate reduction in expression in these tissues (1.5-fold and 3.1-fold reductions in quadriceps and heart RNA, respectively) [23]. The N271D (AAV8 VP1 numbering) variant identified in this study is in VR-1 and in proximity with the previously described residues (Fig 3b), and also shows a profound impact on liver targeting. Together, these converging evidence points to the important role of VR-1 in determining liver targeting. Although the detailed mechanism remains to be elucidated, it may involve the proposed VR-1 hydrogen bond network [24], and/or interactions with AAVR [35], a cellular factor that binds to most AAV capsids and facilitates their cell entry and intracellular trafficking, as structural studies have shown that VR-1 is part of the AAVR-AAV binding footprint [36,37]. As the impact of the N271D mutation on tissue tropism shows a similar trend across all three AAV capsids tested, we postulate that this residue change may exert its effect through modulating the receptor binding. Alternatively, AAV capsid deamidation where an amide group in the side chain of amino acid is converted to a carboxylic acid group, such as N-to-D changes, has been shown to impact transduction efficiency [38]. Further saturating mutagenesis studies will help determine whether the N271D variant exerts its effect through the loss of asparagine (N) or the gain of aspartic acid (D).

Liver toxicity observed in some preclinical and clinical studies has highlighted the need for AAV capsids with liver-detargeting properties, particularly for gene delivery to extrahepatic tissues via systemic administration [39]. Several engineered capsids, such as AAV2i8 [14], AAVMYO [30], MyoAAV [34], PHPeB [40], and VCAP-102 [41], exhibit both enhanced muscle or CNS targeting and modest liver-detargeting (approximately 3- to 30-fold reduction in liver tropism relative to their parental capsids). However, these liver-detargeting effects were typically incidental findings during biodistribution profiling, rather than outcomes of intentional design. Moreover, enhanced targeting of extrahepatic tissues does not necessarily lead to reduced liver-detargeting. For example, the engineered capsid BI-hTFR1 engages the human transferrin receptor to facilitate blood-brain barrier penetration, but retains liver tropism comparable to its parental AAV9 capsid

[42]. Therefore, a modular capsid modification that confers liver-detargeting and can be applied across different capsid backbones represents a promising complementary strategy in capsid engineering. The dramatic liver detargeting phenotype of the AAV8.N271D and MyoAAV.N270D vectors is accompanied by moderate reductions in targeting the heart and TA muscle (Fig 4). A similar trend (i.e., profound liver detargeting with a moderate impact on heart and quadriceps targeting) was also observed when reprogramming AAV9 and Anc80 with the liver toggle residue changes [23]. This creates a dilemma in developing liver-detargeting vectors for gene therapy delivery to muscle tissues. However, the MyoAAV.N270D vector partially retains the excellent muscle targeting property of the parental MyoAAV capsid, and exhibits higher gene transfer and expression levels in the heart and TA muscle than the AAV8 and AAV9 vectors. Furthermore, it may be possible to engineer residues in VR-1, either alone or in combination with changes in other capsid regions, to develop capsids with true liver-specific detargeting capabilities. For example, incorporating the tyrosine mutant Y445F [43] into the T265del AAV6 capsid can further enhance cardiac transduction while maintaining a liver-detargeted biodistribution profile [24].

In summary, this study identifies a naturally occurring AAV capsid variant with a liver detargeting phenotype, reinforces the critical role of VR-1 in modulating liver tropism and transgene expression, and opens new avenues for engineering AAV vectors that minimize liver targeting and associated toxicity.

## Materials and methods

### Ethics statement

All animal studies were reviewed and approved by the Institutional Animal Care and Use Committee of the University of Massachusetts Chan Medical School.

### AAV constructs

The reporter transgene construct used in the small-scale AAV vector production assay contains an *EGFP* transgene under the control of the CMV enhancer and chicken beta-actin promoter. The same construct was packaged in multiple capsids for single vector administration in mice. The barcoded transgene construct used for generating AAV vector library contains a *TuD* transgene under the control of the U6 promoter (also see S1 Fig). The detailed design of barcode has been described previously [29]. Briefly, the barcode contains randomized 8 nucleotides that are embedded in the microRNA binding site (MBS) between stem I and stem II of the *TuD* transgene DNA, and it is also present in the *TuD* RNA following expression. All AAV constructs were designed to package single-stranded vector genomes.

### Small-scale AAV vector production assay

The details of the procedure have been described previously [44]. Briefly, a pair of packaging plasmid and gene-of-interest (GOI) plasmid were co-transfected to HEK293 cells along with a helper plasmid in a 12-well plate using the calcium phosphate method. 72 hours post-transfection, cells and culture media were collected to generate crude lysates following three freeze-and-thaw cycles. The crude lysates were clarified by centrifugation and sequentially treated with DNase I and proteinase K, followed by vector genome determination using droplet digital PCR (ddPCR). To minimize the impact of cell culture condition on rAAV productivity among experimental batches and to reduce variations, we kept the cell density at approximately 80% at transfection and included AAV8 capsid as the normalization control in each batch of experiment. The productivity of each variant was presented as the percentage of the AAV8 productivity.

### Nanopore sequencing of amplicons

AAV vector DNA was extracted with the QIAamp DNA Micro Kit (Qiagen, 56304), and subjected to PCR amplification. The amplicons were purified with the DNA Clean & Concentrator-5 kit (Zymo, D4013) and sequenced with the Oxford Nanopore platform (Plasmidsaurus). Reads were mapped to the reference sequence, demultiplexed by barcodes, and counted using the Geneious Prime software.

## Large-scale AAV vector production

Adherent HEK293 cells were cultured in roller bottles in Dulbecco's Modified Eagle Medium (DMEM) with 10% fetal bovine serum. Helper plasmid, packaging plasmid, and GOI plasmid were co-transfected using the calcium phosphate method. Next day, the cell culture medium was replaced with fresh, serum-free DMEM. Three days post-transfection, cells and culture media were harvested and subjected to purification using two rounds of cesium chloride gradient centrifugation. AAV vectors were dialyzed against phosphate buffered saline and sterilized by passing through a 0.22 μm filter. To generate the pooled rAAV library with a total of $3.6 \times 10^{14}$ vector genomes (vg), we first calculated the amount required for each of the 39 variants to be $9.2 \times 10^{12}$ vg ($3.6 \times 10^{14}$ vg/ 39). Next, based on the titers of the individual vectors, we calculated the vector volumes required to obtain $9.2 \times 10^{12}$ vg, and subsequently mixed the aliquoted vectors accordingly. This resulted in 40.55 mL of pooled rAAV library at the titer of $8.9 \times 10^{12}$ vg/mL ($3.6 \times 10^{14}$ vg/ 40.55 mL). The same library was used for both mouse and ferret screens.

## Animal work

Wildtype, male C57BL/6J mice were treated with AAV vectors via tail vein at six weeks of age, and euthanized four weeks later. For library screen, the dose was $5 \times 10^{11}$ vg/mouse (approximately $2 \times 10^{13}$ vg/kg); for single vector administration, the dose was $3 \times 10^{11}$ vg/mouse. Wildtype ferrets were subjected to anti-AAV8 and anti-AAV9 neutralizing antibody (NAb) screen. Three ferrets (one female and two males) were sero-negative (<1:5 NAb titers), and therefore were used in experiment. They were treated at 10 weeks of age with the AAV vector library via intravenous injection at the dose of $2 \times 10^{13}$ vg/ kg, and euthanized five weeks later.

## Neutralizing antibody assay

The neutralizing antibody (NAb) assay was performed as previously described [45]. Briefly, serum samples were heat-inactivated under 56°C for 35 minutes, and 1:5 diluted in Dulbecco's Modified Eagle's Medium (DMEM) followed by 2-fold serial dilutions. Each diluted serum sample was incubated with $1.2 \times 10^9$ genome copies (GC) of rAAV8.LacZ (for anti-AAV8 assay) or rAAV9.LacZ (for anti-AAV9 assay) under 37°C/5% $CO_2$ for 1 hour. Huh7 cells grown in a 96-well plate were first infected with wildtype adenovirus serotype 5 (100 viral particles per cell). Three to four hours later, 100 μl of pre-incubated rAAV/serum mixture containing $1 \times 10^9$ GC of rAAV.LacZ was added to each well of Huh7 cell culture and incubated under 37°C/5% $CO_2$ for 1 hour. Cells were then cultured with 5% fetal bovine serum under 37°C/5% $CO_2$ overnight. β-galactosidase activity in cell lysate was measured using the Galacto-Star One-Step β-galactosidase Reporter Gene Assay System (Thermo Fisher Scientific, Cat. No. T1014). The transduction inhibition effect of a test sample at each dilution was calculated by comparing with the negative control at the same dilution (i.e., maximal β-galactosidase activity without inhibition). The NAb titer was defined as a range between the lowest dilution factor that yielded more than 50% transduction inhibition and the next dilution factor that could not inhibit transduction by more than 50%.

## Quantification of barcodes by Illumina sequencing

DNA and RNA were isolated from liver samples using the AllPrep DNA/RNA kit (Qiagen, 80204). For muscle tissues, DNA was extracted with the AllPrep DNA/RNA kit (Qiagen, 80204), while RNA was extracted using Trizol. RNA was subjected to treatment with DNase (Qiagen 79254) followed by a clean-up step using the RNA Clean & Concentrator-5 kit (Zymo, R1014). Purified RNA was reverse-transcribed into cDNA using the SuperScript III First-Strand Synthesis System (Thermo Fisher Scientific, 18080051). PCR was performed with barcoded primers and the KOD Hotstart Master Mix (Millipore, 71842) in a total volume of 25 μL that contained 12.5 μL of KOD Hotstart Master Mix, 0.75 μL of forward primer (10 μM), 0.75 μL of reverse primer (10 μM), 50 ng of DNA. Primer binding sites are shown in S1 Fig. PCR cycling condition: 95°C for 2 min, followed by 25 cycles of 95°C/20s, 58°C/10s, and 70°C/10s. PCR amplicons were gel purified and concentrations were determined using a Qubit 3 Fluorometer (Thermo Fisher Scientific). PCR amplicons were pooled at an equal

ratio and sequenced using MiSeq at UMass Chan Deep Sequencing Core. Sequencing data were analyzed using the Galaxy platform [46]. The data were first demultiplexed by the barcodes in PCR primers (S3 Table) using the "barcode splitter" module. This step resulted in datasets representing individual tissue samples. Next, the resulting datasets were demultiplexed by the barcodes in the *TuD* gene (S4 Table), which led to sequencing reads representing their corresponding capsid variants. Barcode and PCR designs are shown in S1 Fig.

To calculate relative vector genome or RNA abundance of each variant in a tissue, its number of reads in a tissue was first divided by that in the library to derive a normalized abundance. The normalized abundance of each variant was then divided by that of AAV8 to derive the relative abundance.

## Quantification of vector DNA and RNA

DNA and RNA were isolated from mouse tissues using the AllPrep DNA/RNA kit (Qiagen, 80204). RNA was reverse-transcribed into cDNA with the High-Capacity cDNA Reverse Transcription kit (Thermo Fisher Scientific, 43-688-13). To quantify vector DNA in the tissues collected from the mice and ferrets treated with the vector library, a duplexing Taqman droplet digital PCR (ddPCR) assay was performed with one reagent targeting stuffer B in the GOI plasmid [forward primer: AGCCCTAGGGATGAACCAGT, reverse primer: AACCCAGGAGTCATTGCATC, probe: AATCTGAGCCACTGAGCCAT, all synthesized by Integrated DNA Technologies (IDT)] and a Taqman reagent targeting mouse *Tfrc* (Thermo Fisher Scientific, 4458367) or ferret *Tfrc* (forward primer: GCTTTAGACCCAGCAGAAGC, reverse primer: GTTCAGCTGCCCCAT-TCTGAG, probe: TCTCCAGCACTCCAGCTGGCA, all synthesized by IDT). To quantify *EGFP* vector DNA in mouse tissues, a duplexing Taqman ddPCR assay was performed with one reagent targeting *EGFP* (Thermo Fisher Scientific, Mr00660654_cn), and the other targeting mouse *Tfrc* (Thermo Fisher Scientific, 4458367). cDNA was quantified using a duplexing Taqman ddPCR assay with the same *EGFP*-targeting Taqman reagent as mentioned above, and a Taqman reagent targeting mouse *Gapdh* (Thermo Fisher Scientific, 4352339E). ddPCR was performed with a QX200 instrument (Bio-Rad) under the cycling condition: 95°C for 10 min, 40 cycles of 94°C/30s and 61°C/60s, followed by 98°C/10min.

## Western blot

Mouse tissues were homogenized using TissueLyser II (Qiagen) in ice-cold T-PER (Thermo Fisher Scientific, 78510) with protease inhibitor (Roche, 4693159001). Total protein concentration in tissue lysate was determined with Pierce BCA Protein Assay Kit (Pierce, 23225). Protein lysates normalized for total protein amount were boiled with 4 × Laemmli sample buffer (Bio-Rad, 1610747) at 99 °C for 5 min. Primary antibodies: mouse anti-EGFP Antibody (Abcam, ab184601, 1:5000), rabbit anti-GAPDH (Abcam, ab9485, 1:10000). Secondary antibodies: LICOR IRDye 680RD goat anti-mouse IgG (H + L) (LI-COR Biosciences, 926–68070, 1:7000), LICOR IRDye 800CW goat anti-rabbit IgG (H + L) (LI-COR Biosciences, 926–32211, 1:7000). Blot membranes were imaged using a LI-COR scanner (Odyssey) and quantified by Li-Cor software.

## Statistical analysis

All statistical analyses were performed using Prism 10 (GraphPad). Two-tailed t-test was used to compare the means of two groups. Two-tailed analysis of variance (ANOVA) was used to compare the means of three or more groups. Two-tailed Mann-Whitney U test was used to compare the medians of two groups.

## Supporting information

**S1 Fig. Barcoded construct and PCR design. (a)** Schematics showing the barcoded AAV construct, not drawn to scale. The first barcode (bc1) in the TuD and the second barcode (bc2) in the forward PCR primer are rainbow-colored. Note that each unique bc1 was used to pair with a unique capsid variant during rAAV production, and that bc2 was used in PCR

to differentiate among tissue samples. **(b)** The DNA sequence of the construct shown in a. Key elements are color-coded as the scheme shown in a. Gray: ITR. Blue: stuffer. Green: U6 promoter. Yellow: TuD. NNNNNNNN: bc1. **(c)** Binding sites of the forward and reverse PCR primers shown in a. Binding sites are underlined. Note that bc2 is included at the 5' end of the forward primer.
(PDF)

**S2 Fig. Validation of the relative distribution of barcoded PCR amplicons derived from vector library DNA. (a)** Bar graph showing the count of Illumia sequencing (MiSeq) reads mapped to the unique vector transgene barcodes packaged in AAV8, AAV9, or AAV8 variants. Data were based on one biological repeat. **(b)** Scatter dot plot showing the relationship between the sequencing read counts by nanopore method (x-axis) and MiSeq method (y-axis). The linear regression statistics are shown.
(PDF)

**S3 Fig. Tabulation of neutralizing antibody titers in ferret serum samples.** Serum samples were collected immediately prior to rAAV injection (at rAAV injection) or immediately prior to euthanasia (at euthanasia). M: male. F: female.
(PDF)

**S4 Fig. Vector genome abundance in mouse and ferret tissues.** Scatter dot plot showing the vector genome copy number per diploid host genome (vg/diploid genome) in the liver, heart, and tibialis anterior (TA) muscle collected from the mice (blue dots) and ferrets (green dots) treated with the vector library. Each dot represents an individual animal. Data are shown as mean and standard deviation. Statistical analysis is performed using two-tailed t-test.
(PDF)

**S5 Fig. Workflow to select AAV8 variants.**
(PDF)

**S6 Fig. Vector titers by large-scale production.** The parental capsid of each vector pair is indicated below the bars. The mutant capsids are AAV8.N271 in the AAV8 pair, AAV9.N270D in the AAV9 pair, and MyoAAV.N270D in the MyoAAV pair. All vectors carry the same vector genome that expresses EGFP.
(PDF)

**S7 Fig. Western blotting to quantify transgene expression. (a-c)** EGFP (green) and GAPDH (red) signals in the liver, heart, and tibialis anterior (TA) muscle tissue lysates. Mice were treated with AAV8 or AAV8.N271D vectors (a), AAV9 or AAV9.N270D vectors (b), and MyoAAV or MyoAAV.N270D vectors (c). Mice treated with PBS serve as negative controls. Different treatment groups are separated by dashed white lines to enhance visualization. Each lane represents an individual mouse. **(d)** Comparison among AAV8, AAV9, and MyoAAV.N270D vectors by running relevant samples on the same gels. Quantification is shown on the right side. Each dot represents an individual mouse. The box extends from the first to the third quartiles with the line inside denoting median. The whiskers end at minimum and maximum values. The fold changes of medians and p values are labeled. Statistical analysis is performed using two-tailed non-parametric Mann-Whitney U test.
(PDF)

**S8 Fig. Relative gene delivery and transgene expression.** The levels in liver relative to those in heart (a) and TA muscle (b) in individual mice are calculated using the data shown in Fig 4 and plotted. Each dot represents an individual mouse. The box extends from the first to the third quartiles with the line inside denoting median. The whiskers end at minimum and maximum values. The fold changes of medians and p values are labeled. Statistical analysis is performed using two-tailed non-parametric Mann-Whitney U test.
(PDF)

**S1 Table. Amino acid sequences of AAV8 capsid variants.**
(XLSX)

**S2 Table. Unique residues among AAV8 capsid variants.**
(XLSX)

**S3 Table. Barcodes in the forward PCR primers (bc2).**
(XLSX)

**S4 Table. Barcodes in the TuD genes (bc1).**
(XLSX)

## Author contributions

**Conceptualization:** Ruxiao Xing, Dan Wang.

**Data curation:** Ruxiao Xing, Mengyao Xu, Darcy Reil, Alisha M Gruntman, Dan Wang.

**Formal analysis:** Ruxiao Xing, Mengyao Xu, Darcy Reil, Alisha M Gruntman, Dan Wang.

**Funding acquisition:** Terence R Flotte, Guangping Gao, Dan Wang.

**Investigation:** Ruxiao Xing, Mengyao Xu, Darcy Reil, April Destefano, Nan Liu, Jialing Liang, Meiyu Xu, Fang Zhang, Alisha M Gruntman.

**Methodology:** Ruxiao Xing, Mengyao Xu, Darcy Reil, April Destefano, Mengtian Cui, Nan Liu, Jialing Liang, Meiyu Xu, Fang Zhang, Phillip WL Tai, Alisha M Gruntman.

**Project administration:** Dan Wang.

**Resources:** Mengtian Cui, Guangchao Xu, Li Luo, Fang Zhang, Phillip WL Tai, Yuquan Wei, Alisha M Gruntman, Terence R Flotte.

**Software:** Mengtian Cui, Phillip WL Tai.

**Supervision:** Guangping Gao, Dan Wang.

**Validation:** Ruxiao Xing, Mengyao Xu.

**Visualization:** Ruxiao Xing, Mengyao Xu, Dan Wang.

**Writing – original draft:** Mengyao Xu.

**Writing – review & editing:** Dan Wang.

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
