## [Decision Letter · Decision Letter 0]

23 Apr 2025

PPATHOGENS-D-25-00584

A single amino acid variant in the variable region I of AAV capsid confers liver detargeting

PLOS Pathogens

Dear Dr. Wang,

Thank you for submitting your manuscript to PLOS Pathogens. After careful consideration, we feel that it has merit but does not fully meet PLOS Pathogens's publication criteria as it currently stands. Therefore, we invite you to submit a revised version of the manuscript that addresses the points raised during the review process.

Please submit your revised manuscript within 60 days Jun 22 2025 11:59PM. If you will need more time than this to complete your revisions, please reply to this message or contact the journal office at plospathogens@plos.org. Please include the following items when submitting your revised manuscript:

We look forward to receiving your revised manuscript.

Kind regards,

WEIDONG XIAO

Guest Editor

PLOS Pathogens

Alison McBride

Section Editor

PLOS Pathogens Sumita Bhaduri-McIntosh

Editor-in-Chief

PLOS Pathogens

orcid.org/0000-0003-2946-9497

 Michael Malim

Editor-in-Chief

PLOS Pathogens

orcid.org/0000-0002-7699-2064

**Additional Editor Comments :**

The reviewers have provided valuable opinions on your studies. Please address their concerns in your revision.

**Journal Requirements:**

At this stage, the following Authors/Authors require contributions: Ruxiao Xing, Mengyao Xu, Darcy Reil, April Destefano, Mengtian Cui, Nan Liu, Jialing Liang, Guangchao Xu, Li Luo, Meiyu Xu, Fang Zhang, Phillip WL Tai, Alisha M Gruntman, Terence R Flotte, Guangping Gao, and Dan Wang. Please ensure that the full contributions of each author are acknowledged in the "Add/Edit/Remove Authors" section of our submission form.

https://journals.plos.org/plospathogens/s/submission-guidelines#loc-parts-of-a-submission

5) We notice that your supplementary Figure is included in the manuscript file. Please remove it and upload it with the file type 'Supporting Information'. Please ensure that each Supporting Information file has a legend listed in the manuscript after the references list.

6) We note that your Data Availability Statement is currently as follows: "All data are included in this manuscript.". Please confirm at this time whether or not your submission contains all raw data required to replicate the results of your study. Authors must share the “minimal data set” for their submission. PLOS defines the minimal data set to consist of the data required to replicate all study findings reported in the article, as well as related metadata and methods (https://journals.plos.org/plosone/s/data-availability#loc-minimal-data-set-definition).

7) Please provide a completed 'Competing Interests' statement, including any COIs declared by your co-authors. If you have no competing interests to declare, please state "The authors have declared that no competing interests exist". Otherwise please declare all competing interests beginning with the statement "I have read the journal's policy and the authors of this manuscript have the following competing interests:"

**Reviewers' Comments:**

Reviewer's Responses to Questions

**Part I - Summary**

Reviewer #1: The manuscript investigates the role of single variable regions (VR) in the tropism of AAV, focus on VR1 structure. Here, Wang group found the N271D mutation could lead to liver de-targeting in mouse model. This topic is interesting and highlights a promising approach for engineering AAV vectors. The data presented are supportive.

Reviewer #2: In this study from the renowned Wang and Gao labs, Xing and colleagues harnessed their availability of a vast collection of AAV capsid variants found directly in human clinical samples to generate a focused library of 159 AAV8 variants, which was then screened in mice and ferrets. Strikingly, this identified a single residue (N271D) that determined liver (de-)targeting in both species. While the concept of single-residue liver toggles has been reported before (as correctly referenced by the authors), this study is still important because 1) it identifies yet another critical residue with this property, 2) it does so using a unique and original approach, and 3) it expands on our understanding of the critical role of variable region 1 in AAV for liver (de-)targeting. This teaches the field about AAV virus and vector biology, and thus extends our capabilities to rationally engineer better and safer tools for human gene therapy.

Reviewer #3: In this manuscript, Xing et al. report a study in which they performed a screening of AAV8 variants and identified N271D as a liver detargeting mutation that retains the ability to transduce heart and skeletal muscle. The authors initially screened 159 AAV8 variants for AAV vector yields and selected the top 37 candidates for barcode-based in vivo evaluation in mice and ferrets aiming at identifying variants that outperform AAV8 for liver transduction. However, unfortunately, none of the 37 variants were found to be superior to AAV8 in liver transduction. Instead, they identified 8 AAV8 variants with a liver-detargeting phenotype. Among these, 5 variants retained the ability to transduce heart and skeletal muscle, albeit at varying reduced levels, including an AAV variant carrying N271D mutation. The authors then investigated whether the N271D's liver-detargeting phenotype could be transferred to other capsids, AAV9 and MyoAAV, without significantly compromising their transduction efficiency in heart and skeletal muscle, only yielding modest success.

Overall the study appears to be a loosely assembled collection of experiments lacking a well-justified cohesive scientific rationale, not providing any noteworthy impacts in the field. Rigor in overall work is lacking and there are major concerns as summarized below.

**Part II – Major Issues: Key Experiments Required for Acceptance**

Reviewer #1: The following questions and comments should be addressed to strengthen the study.

1. Introduction requires a restructure to emphasize the importance and functions of VR, especially in tropism. Please include the concise overview of the VR1 structure features and their corresponding impact on tropism across all AAV clade, previous related mutations or design strategies on altering liver tropism.

2. Clarify the purpose of the AAV library: The rationale behind generating and characterizing an AAV vector library should be explicitly stated early in the manuscript. How does this library approach advance the field compared to rational design or directed evolution?

3. Justification for N271D and selection of AAV5: is the N271D the common / conserved variant among v2, v5, v23, v25, v35, and V12, v13, v15? phylogenetic or structural alignment analysis would strengthen the argument for focusing on this residue.

4. Enhance discussion on liver de-Targeting significance: Compare with other known liver-detargeting mutations?

5. More discussion on N vs. D functional impact: given the different chemical classification and function, the contribution of capsid on the structure and function, between N and D, how will the swap alter the capsid structure and receptor binding? More discussion needs to be strengthened. And additional types of mutation on this site are encouraged to be further tested on the mechanistic basis ….

6. s1a: How many samples were included in PBS group? The absence of loading control bands in the PBS group raises concerns about data reliability. Please address this.

Reviewer #2: This work is very straightforward, clear and conclusive, thus my only question is whether it would be possible and make sense to add a graph showing the date for heart:liver and TA:liver for selected capsids? Thus far, the authors have normalized their data to AAV8 in each tissue, but it could help to further illustrate the improved relative liver detargeting if vector abundance and mRNA expression in the muscle tissues are normalized to the corresponding data in the liver for each variant (and the AAV8 benchmark, for which these ratios are presumably worse with respect to liver detargeting)?

Reviewer #3: Many liver-detargeting capsid amino acid modifications have already been reported in prior studies by other groups and some of those have been shown to retain important attributes and be transferable to other AAV capsids. The data presented in Figure 4 indicate that the N270D/N271D mutation does not provide any compelling evidence of effective transferability with no loss of key attributes of capsids. Rather the data demonstrate that carrying the N270D/N271D mutation in non AAV8 capsids leads to substantial and statistically significant loss of transgene expression levels in heart and skeletal muscle, ranging from 4- to 16-fold reduction compared to the parental capsids. Thus, these findings are unlikely to have a meaningful impact on the field.

The overall goal of the study is unclear. Both the abstract and the introduction fail to clearly articulate the purpose of the study and the important outcomes in light of the aim of the study. The manuscript focuses primarily on a single incidental finding that does not provide new insights or represent a meaningful advancement in the field.

The justification of the experimental design and key methodological details are either absent or insufficiently described. Examples include the following. The descriptions on how the 159 AAV8 variants were identified and their sequences are missing. The selection criteria for AAV8 variants and their subgroups comprising 37, 8 and 5 AAV8 variants are ambiguous. Critical information regarding plasmid construction, barcode design, NGS sequencing and barcode data analysis methods, and qPCR procedures including primer sequences and PCR conditions is missing or inadequately described. The rationale for choosing ferrets as the target animal model and for using two females and one male is not provided. Some analyses seem to be incomplete. For example, genome transcripts levels in liver and other organs were assessed in mice but were not analyzed in ferrets. The rationale for the selection of MyoAAV in the study is not described. Details on sample sizes and replicates are missing in several experiments (e.g., Fig. 1).

No statistical analysis of the NGS data was performed. Rigorous statistical analyses are essential for making conclusions.

" All data are included in this manuscript" is not correct. The authors need to make all data available to the readership including the NGS data.

"We extracted the pooled library vector DNA, amplified the region with barcodes by PCR, and conducted high-throughput nanopore sequencing to quantify the relative abundance of each barcode, which in turn reflected the relative abundance of each capsid vector in the library." This statement is misleading. Without normalization by input read counts, output read counts by themselves cannot reliably represent relative abundance of each AAV capsid because variations in barcode sequences can introduce amplification biases.

Thus, Fig. 1b, Y-axis does not necessarily reflect relative vector abundance.

The statement, "In contrast, purifying all capsid vectors in bulk would likely result in skewed representation in favor of good producers, which may cause bias in subsequent functional screens," may not necessarily correct. As long as individual capsid titers are adjusted at the crude lysate stage prior to pooling for bulk purification, such bias toward high-yielding variants can be effectively minimized.

**Part III – Minor Issues: Editorial and Data Presentation Modifications**

Reviewer #1: (No Response)

Reviewer #2: None

Reviewer #3: Since the study used both nanopore and Illumina sequencing, the authors need to specify what NGS was used in the main text. ".... quantified by NGS..." is an ambiguous statement.

Fig. 4. All capsids should be presented within a single graph to allow for direct comparison, and statistical analyses should be conducted across the entire dataset. In particular, it is important to statistically compare all AAV9-derived capsids, the wild-type AAV9 and the three AAV9 variants.

PLOS authors have the option to publish the peer review history of their article (what does this mean? ). If published, this will include your full peer review and any attached files.

**Do you want your identity to be public for this peer review?** For information about this choice, including consent withdrawal, please see our Privacy Policy .

Reviewer #1: No

Reviewer #2: No

Reviewer #3: No

**Figure resubmission:**
---

## [Decision Letter · Decision Letter 1]

5 Aug 2025

PPATHOGENS-D-25-00584R1

A single amino acid variant in the variable region I of AAV capsid confers liver detargeting

PLOS Pathogens

Dear Dr. Wang,

Thank you for submitting your manuscript to PLOS Pathogens. After careful consideration, we feel that it has merit but does not fully meet PLOS Pathogens's publication criteria as it currently stands. Therefore, we invite you to submit a revised version of the manuscript that addresses the points raised during the review process.

Please submit your revised manuscript within 30 days Oct 04 2025 11:59PM. If you will need more time than this to complete your revisions, please reply to this message or contact the journal office at plospathogens@plos.org. Please include the following items when submitting your revised manuscript:

We look forward to receiving your revised manuscript.

Kind regards,

WEIDONG XIAO

Guest Editor

PLOS Pathogens

Alison McBride

Section Editor

PLOS Pathogens

Sumita Bhaduri-McIntosh

Editor-in-Chief

PLOS Pathogens

orcid.org/0000-0003-2946-9497

Michael Malim

Editor-in-Chief

PLOS Pathogens

orcid.org/0000-0002-7699-2064

**Additional Editor Comments:**

All three reviewers have raised additional questions which should be addressed. Reviewer 3's expert opinions should be carefully considered in your revision.

**Journal Requirements:**

Please ensure that the funders and grant numbers match between the Financial Disclosure field and the Funding Information tab in your submission form. Note that the funders must be provided in the same order in both places as well.

State what role the funders took in the study. If the funders had no role in your study, please state: "The funders had no role in study design, data collection and analysis, decision to publish, or preparation of the manuscript.".

**Reviewers' Comments:**

Reviewer's Responses to Questions

**Part I - Summary**

Reviewer #1: Revision significantly improved the data description and explanation, which is easier to follow.

Reviewer #2: The authors have addressed all my previous concerns, hence there is nothing to add here.

Reviewer #3: Xing et al. revised the manuscript in response to the critiques raised by three reviewers. They were very responsive and most of the revisions seem appropriate. The data appear scientifically sound. That being said, the major concern that significantly diminishes the scientific merit of the paper remains unresolved, that is, whether the study findings will have a significant impact on the field. If the authors had identified transferable VR-I liver-detargeting mutations that could mediate transduction in non-hepatic target organs (e.g., heart and skeletal muscle) at levels equivalent to or higher than those of the parental AAV capsids, the study would likely warrant a high impact. However, the manuscript instead merely highlights a single, serendipitously identified mutation that reduces liver tropism while retaining the ability to transduce non-hepatic organs at moderately to substantially lower levels (4- to 16-fold) when introduced into capsids other than the one in which the phenotype was initially identified. Many similar liver-detargeting variants have already been reported in the literature, granted patents, and published patent applications. Importantly, the key claim that the liver-detargeted MyoAAV.N270D can mediate superior transduction in the heart and TA muscle compared to AAV8 and AAV9 (Lines 178-180) is not convincingly demonstrated by the data presented in the manuscript (Right panels in Fig. 4). Therefore, although the reported liver-detargeting mutation might be a useful addition to the growing list of such mutations, its discovery lacking a novel and compelling phenotype is unlikely to represent a significant advancement in the field.

**Part II – Major Issues: Key Experiments Required for Acceptance**

Reviewer #1: all questions have been answered.

Reviewer #2: None

Reviewer #3: Lines 178-180, the authors’ argument, “Although MyoAAV.N270D.EGFP led to lower transduction levels in the heart and TA muscle as compared to MyoAAV.EGFP, it still outperformed AAV9.EGFP and AAV8.EGFP,” is not at all supported by the data presented. A cursory analysis for protein expression levels by eyes (Right panels in Fig. 4) revealed: MyoAAV.N270D=~0.1, AAV9=~1.0, and AAV8=~1.0 in the heart, and MyoAAV.N270D=~0.2, AAV9=~1.0, and AAV8=~1.0 in the TA muscle. This data does not align with the authors’ argument, and rather shows that AAV8 and AAV9 outperform MyoAAV.N270D. It appears that, to support their preferred claim, the authors focused only on mRNA expression levels while dismissing protein expression levels, which represent a more relevant and direct measure of transduction efficiency. In addition, the authors did not perform a statistical analysis to compare transduction levels in the heart and TA muscle between MyoAAV.N270D, AAV9, and AAV8 (i.e., a three-way comparison). The authors should analyze the data more rigorously and objectively to make a conclusion.

Line 112, “with the potential caveat that different barcode sequences may introduce PCR bias.” This effect is often substantial. How did the authors address this caveat when analyzing and interpreting the data? This information is missing.

Please add a "Statistical analysis" paragraph in the Materials and Methods section describing the statistical methods used for data comparisons. In addition, in each figure legend, please describe briefly how the data were statistically analyzed, including the tests performed and any corrections made for multiple comparisons, if applicable. Moreover, please indicate which type of test, one-tailed or two-tailed, was used.

**Part III – Minor Issues: Editorial and Data Presentation Modifications**

Reviewer #1: Fig1 a: tittle for x axis is suggested to be added to facilitate the data interpretation. Please state how AAV yield from each variant was compared and how comparisons were made to ensure fairness? Was any internal control used to normalize transfection efficiency or yield? It is noteworthy that certain mutations result in a dramatic decrease in yield—additional explanation or hypotheses would be valuable. Was the mutation in v2, 5, 12, 15, 23, 25, 36 significantly effected in yield with AAV8, AAV9 and Myo?

Line124: Please specify the method used to generate the pooled AAV preparation from single mutation and how the viral genome (vg) titer was calculated to achieve the final dosing concentration of 2 × 10¹³ vg/kg for injection.

Line 152: Among so many interesting single mutation variants, N271D was selected for further study. Please rephrase this sentence to explicitly state the rationale for a smooth transition in the context.

Discussion: It would strengthen the discussion to further elaborate on how liver de-targeting by the N271D mutation appears to be independent of AAV serotype, as similar trends are observed across AAV8, AAV9, and Myo capsids. Expanding on this point would provide insight beyond the role of capsid deamidation alone.

Reviewer #2: None

Reviewer #3: Lines 59-60. The authors are encouraged to double-check whether the statement, “Indeed, VR-1 contributes to the 3-fold protrusions on AAV capsid”, is appropriate.

Fig. 1a. To improve clarify, please provide information on how each point corresponds to the AAV8 variants listed in Table S1.

Lines 135-136, the statement, “low AAV vector DNA abundance in tissues does not necessarily result in low functional transduction,” is misleading. This phenomenon is more relevant when an organ carries a relatively high AAV vector genome copy number but exhibits disproportionately low transgene expression, rather than the opposite scenario, where vector genome copy number is low but transgene expression is disproportionately high.

Line 180, “middles” is a typo. It should be “middle.”

Lines 217-219, regarding the following statement, "The dramatic liver detargeting phenotype of the AAV8.N271D, AAV9.N270D, and MyoAAV.N270D vectors is accompanied by moderate reductions in targeting the heart and TA muscle (Figure 4)." Since AAV9.N270D exhibits 15- to 16-fold (substantial) reduction in transgene expression in the heart, "moderate" in the above statement is misleading. Please revise this statement appropriately (e.g., by removing AAV9.N270D).

Lines 301-306. The authors need to provide a brief description explaining how raw read count numbers were converted to relative values presented in the figures, which is missing in the manuscript.

Line 400, "Mann Whitney test" should be "Mann-Whitney U test."

What is the justification for using a t-test for some datasets and a U test for others?

PLOS authors have the option to publish the peer review history of their article (what does this mean? ). If published, this will include your full peer review and any attached files.

**Do you want your identity to be public for this peer review?** For information about this choice, including consent withdrawal, please see our Privacy Policy .

Reviewer #1: No

Reviewer #2: No

Reviewer #3: No

**Figure resubmission:**
---

## [Editor Report · Decision Letter 2]

9 Sep 2025

Dear Dr. Wang,

We are pleased to inform you that your manuscript 'A single amino acid variant in the variable region I of AAV capsid confers liver detargeting' has been provisionally accepted for publication in PLOS Pathogens.

Best regards,

WEIDONG XIAO

Guest Editor

PLOS Pathogens

Alison McBride

Section Editor

PLOS Pathogens

Sumita Bhaduri-McIntosh

Editor-in-Chief

PLOS Pathogens

orcid.org/0000-0003-2946-9497

Michael Malim

Editor-in-Chief

PLOS Pathogens

orcid.org/0000-0002-7699-2064

The revision is extensive and thorough. I recommend the manuscript for publication.
---

## [Editor Report · Acceptance letter]

Dear Dr. Wang,

We are delighted to inform you that your manuscript, "A single amino acid variant in the variable region I of AAV capsid confers liver detargeting," has been formally accepted for publication in PLOS Pathogens.

Best regards,

Sumita Bhaduri-McIntosh

Editor-in-Chief

PLOS Pathogens

orcid.org/0000-0003-2946-9497

Michael Malim

Editor-in-Chief

PLOS Pathogens

orcid.org/0000-0002-7699-2064